# Barriers and facilitators to uptake and retention of inner-city ethnically diverse women in a postnatal weight management intervention: a mixed-methods process evaluation within a feasibility trial in England

Cath Taylor ,[1] Vanita Bhavnani,[2] Magdalena Zasada,[1] Michael Ussher,[3,4] Debra Bick ,[5] SWAN trial team, On behalf of the SWAN trial team

For numbered affiliations see end of article.

**Correspondence to**
Cath Taylor;
cath.taylor@surrey.ac.uk

## ABSTRACT

**Objectives** To understand the barriers and facilitators to uptake and retention of postnatal women randomised to a commercial group weight management intervention using the COM-B (capability, opportunity, motivation and behaviour) behaviour change model.

**Design** Concurrent mixed-methods (qualitative dominant) process evaluation nested within a feasibility randomised controlled trial, comprising questionnaires and interviews at 6 and 12 months postbirth.

**Setting** One National Health Service maternity unit in an inner city area in the south of England.

**Participants** 98 postnatal women with body mass indices>25 kg/m[2] (overweight/obese) at pregnancy commencement.

**Intervention** Twelve-week Slimming World (SW) commercial group weight management programme, commencing anytime from 8 to 16 weeks postnatally.

**Primary and secondary outcome measures** Data regarding uptake and retention from questionnaires and interviews conducted 6 and 12 months postbirth analysed thematically and mapped to the COM-B model.

**Results** Barriers to SW uptake mostly concerned opportunity issues (eg, lack of time or childcare support) though some women also lacked motivation, not feeling that weight reduction was a priority, and a few cited capability issues such as lacking confidence. Weight loss aspirations were also a key factor explaining retention, as were social opportunity issues, particularly in relation to factors such as the extent of group identity and relationship with the group consultant; and physical opportunity such as perceived support from and fit with family lifestyle. In addition, barriers relating to beliefs and expectations about the SW programme were identified, including concerns regarding compatibility with breastfeeding and importance of exercise. Women's understanding of the SW approach, and capability to implement into their lifestyles, appeared related to level of attendance (dose–response effect).

**Conclusions** Uptake and retention in commercial weight management programmes may be enhanced by applying

behaviour change techniques to address the barriers impacting on women's perceived capability, motivation and opportunity to participate.

**Trial registration number** ISRCTN39186148.

## Strengths and limitations of this study

► This is this first study to use behaviour change theory to explore uptake and retention of postnatal women in a weight management intervention.
► The sample were from an ethnically and economically diverse inner-city population.
► The intervention is an evidence-based commercially available programme available throughout the UK and in several other countries worldwide.
► As a single-centre feasibility study, findings cannot necessarily be generalised.
► Techniques proposed to mitigate the emergent barriers require further evaluation.

## INTRODUCTION

The proportion of women entering pregnancy with an overweight or obese body mass index (ie, BMIs of ≥25 kg/m[2]) is high and increasing, accounting for approximately half of all pregnancies in the UK[1] and USA.[2] Pregnant women with obese BMIs have an increased risk of adverse outcomes for themselves (eg, pre-eclampsia, gestational diabetes) and their infants (eg, stillbirth, congenital anomalies).[3] Furthermore, retaining weight postnatally is associated with other poor health behaviours including smoking, poor diet, lack of exercise and not breastfeeding.[4–6] It also increases risks for long-term obesity, hypertension, diabetes and degenerative joint disease in women.[7] The infants of women with obese BMIs in

pregnancy are also at risk of higher BMI and blood pressure themselves in childhood and young adulthood.[6]

Intervening prior to pregnancy is challenging as most pregnancies are unplanned[8] and women are not generally in touch with health services until they are pregnant. Due to limited evidence of effective weight management interventions during pregnancy,[9] attention has shifted to the postnatal period. However, studies of postnatal weight management have methodological limitations (eg, lacking information on randomisation or response rates), and wide heterogeneity in relation to intervention and evaluation.[10 11] Furthermore, neither uptake nor retention are explicitly reported in these reviews.

In general populations, commercial weight management programmes were more effective than health service delivered programmes,[12] with the advantage of being nationally available and tailored to community needs and settings. Few cater for women in pregnancy or postpartum; one exception is Slimming World (SW, www.slimmingworld.co.uk) who have produced guidelines and support in collaboration with the Royal College of Midwifery.

According to the COM-B (capability, opportunity, motivation and behaviour) model,[13] successful behaviour change, such as weight management, requires an individual to be *capable* of performing the behaviour, have the *opportunity* to carry out the behaviour and be *motivated* to change the behaviour. For general populations, COM-B has been applied in a review of uptake and retention in weight management programmes (including SW).[14] Uptake was greatest in those motivated to attend and this was highly related to capability (knowledge and skills to change behaviour) and opportunity, particularly social opportunity. Social opportunity (including within the programme; eg social support, supportive leader and outside; eg, family and friends) was the dominant driver of retention, with psychological capability also important (highest retention when there was self-monitoring and feedback on behaviour). A recently completed feasibility randomised controlled trial of access to SW in a postnatal population[15] presented data on recruitment, retention and other feasibility outcomes, which were supportive of progression to a definitive trial. This paper reports results from the mixed-methods process evaluation of the study, the objective being to use the COM-B to examine the barriers and facilitators to uptake and retention of women allocated to the intervention.

## METHODS

Reported according to Consolidated criteria for Reporting Qualitative research guidelines.

### Design

Concurrent mixed-methods (qualitative dominant) process evaluation nested within a feasibility randomised controlled trial, comprising questionnaires and interviews at 6 and 12 months postbirth. The study was conducted between 1 December 2015 and 30 November 2018.

### Setting

One National Health Service (NHS) maternity unit in an ethnically diverse inner city area in the south of England with high levels of socioeconomic disadvantage, mobility and migration.

### Participants

Postnatal women aged ≥18 with BMIs≥25 kg/m$^2$ at pregnancy commencement or with normal BMIs (18.5–24.9kg/m$^2$) who met criteria for excessive gestational weight gain (EGWG) at 36 weeks gestation, randomly allocated at 36 weeks gestation to access 12-weekly SW sessions, commenced anytime from 8 to 16 weeks postbirth (n=98). Full inclusion/exclusion criteria are published elsewhere.[15]

### Intervention

SW (Alfreton, UK) is a commercial weight management programme. The programme is underpinned by behaviour change theory and techniques, including social cognitive theory[16] with a focus on motivation and self-efficacy for weight management and reducing relapse. Key techniques include goal setting, self-monitoring, social support and positive reinforcement.[17 18] Consultants leading the groups receive standardised training from SW dieticians and nutritionists, which includes motivational skills to support positive lifestyle changes, nutrition, food facts and role of exercise and activity in health and weight management. Groups follow a standard format, starting with members being weighed (confidentially), while new members attend an introductory session, followed by group session (called image therapy) where members discuss their experiences of weight management that week, facilitated by the consultant and aimed at supporting behaviour change, and encouraging peer support. Sessions can include basic cooking skills, taking cost, cultural preferences and time constraints into account. Physical activity is encouraged and rewarded through 'body magic awards', including redefining what 'activity' can include (eg, housework, gardening). Members also have access to SW online resources (weight tracking, recipes, tips, motivational articles, etc). The dietary plan encourages most foods (80%) to be fruit, vegetables, carbohydrates and lean protein; a smaller but 'mandatory' allowance for calcium-rich and fibre-rich foods; and a daily optional allowance for foods high in fat or sugar. Women who are breastfeeding have an increased allowance for calcium-rich foods.

Women allocated to the intervention were offered (fees waived) the standard SW membership (12 sessions over 14 weeks). SW guidance is that in order to achieve a 5% weight loss, a difference considered to improve health outcomes, individuals need to attend at least 10 of 12 sessions.[19] In our process evaluation, we considered

whether women stayed for whole group sessions (or left after being weighed), and whether women attended 10+sessions, 6–9 sessions; 1–5 sessions or did not attend at all. On completion of the 12 weeks, women could continue with standard SW fees applicable. Women received an information leaflet about joining SW, a congratulations text from the Research Midwives on the birth of their baby, and reminder texts at 6–8 weeks postnatally. From 8 to 16 weeks postnatally, they could call SW to register and locate their nearest groups.

### Questionnaire component

All women were invited to complete questionnaires at baseline (36 weeks gestation) and 6 months postbirth. Responders at 6 months were invited to complete a further questionnaire at 12 months postbirth. The 6- and 12-month questionnaires, which included measures of dietary intake, mental and physical health and life-style behaviours (reported elsewhere)[15], also assessed attendance at SW. This included uptake (asking if they attended any sessions and, if so, age of their baby when attending their first session); and retention (how many sessions attended in total; if they stayed for the whole session or just got weighed). Two open questions asked why they did not stay for whole sessions (if relevant), and how useful they found the sessions. Women who had not attended any sessions were asked to explain why they chose not to attend.

### Interview component

Interviews were conducted by a female experienced qualitative researcher (VB) with a purposively selected sample of women at approximately 6 months postbirth and 12 months postbirth. Criteria for inviting women captured diversity (in line with guidance[20]) in relation to: attendance at SW sessions (women who attended 10+sessions vs attended fewer or did not attend at all); weight change and ethnicity. The same criteria were applied at 12 months postbirth, but also included six women from the initial sample to provide a nested longitudinal sample. The topic guide included motivation for study participation, experiences of the intervention and reasons for uptake and retention (or not), with prompts informed by the COM-B model. All interviews were by telephone (face-to-face was offered), and lasted between 20 and 50 min (average 45 min). Interviews were audiorecorded. Barriers and facilitators to study participation were considered separately to those that relate to the intervention and are reported elsewhere.[15]

### Analysis

Quantitative questionnaire data were analysed descriptively. Analysis of open questions and interview data was underpinned by the COM-B model[13] and used the Framework method.[21] Interviews were transcribed verbatim. Transcripts were read by two researchers (CT and MZ for questionnaires; CT and VB for interviews), who independently noted key themes. Coding was compared and discussed before final coding and labelling. Data were examined to identify themes in relation to capability, motivation and opportunity factors (eg, comparing women who attended different 'doses' of sessions; and those that lost or gained weight). Data were examined longitudinally if women completed both questionnaires/interviews. As the COM-B components are not mutually exclusive, data often fit several dimensions.

### Integration of findings

Data from questionnaires and interviews were analysed separately before being compared, and integrated into a single framework. Key barriers to uptake and retention were mapped to behaviour change techniques (BCT) for enhancing uptake and retention.

Quotations are anonymised with pseudonyms (flowers). Each quote is labelled with the pseudonym, number of sessions attended and weight change from baseline to the time point they provided data.

### Patient and public involvement

The choice of SW as the intervention was heavily influenced by feedback from local women who had participated in a trial of weight management during pregnancy.[22] Their views included that the intervention should be non-NHS, peer supported to enable social support but not just 'new mums' (though knowledgeable of their needs), flexible and suitable for taking their babies. Four women who had recently given birth and experienced weight management issues were recruited to join our PPI group via advertisements in community midwifery locations. They were supported by a researcher (VB) to contribute to the development of study materials, conduct of the trial, interpretation and dissemination of findings.

## RESULTS

Of 98 women randomised to the intervention, 83 (85%) completed the 6-month questionnaire, 69 of whom (69/83, 83%) also completed the 12-month questionnaire. At least one of the two questions about SW was answered by all women who attended at least one SW session. Fifty-two (53%) women did not attend any sessions and 39 (75%) gave a reason for not attending.

Interviews were conducted with 13 women immediately post-intervention (approximately 6–8 months postbirth); and with nine women (six of whom had participated at 6 months) at 12 months postbirth. The characteristics of participants are provided in table 1.

### Uptake

Just over half of the intervention women did not attend any SW sessions. Most commonly this related to 'opportunity', particularly lack of time to attend. Some had difficulties organising childcare or fitting SW sessions into their family schedule, for example sessions clashing with older children's bedtimes. One woman reported breastfeeding on demand as a limiting factor, either

**Table 1** Characteristics of study participants

| Characteristics | Intervention group n=98 | Survey response n=83* | Interview 6 months n=13 | Interview 12 months n=9 |
|---|---|---|---|---|
| **Ethnicity** | | | | |
| White | 49 (50%) | 43 (52%) | 10 (77%) | 6 (67%) |
| Black | 35 (36%) | 27 (33%) | 3 (23%) | 3 (33%) |
| Asian | 8 (8%) | 8 (10%) | 0 (0%) | 0 (0%) |
| Other | 6 (6%) | 3 (4%) | 0 (0%) | 0 (0%) |
| Unknown | 0 (0%) | 2 (2%) | 0 (0%) | 0 (0%) |
| **Age** | | | | |
| 20–24 | 7 (7%) | 4 (5%) | 1 (8%) | 0 (0%) |
| 25–29 | 19 (19%) | 17 (20%) | 3 (23%) | 1 (11%) |
| 30–34 | 33 (34%) | 26 (31%) | 4 (31%) | 3 (33%) |
| 35–39 | 33 (34%) | 30 (36%) | 4 (31%) | 4 (44%) |
| 40–44 | 6 (6%) | 6 (7%) | 1 (8%) | 1 (11%) |
| **BMI at booking** | | | | |
| <25 with EGWG | 3 (3%) | 3 (4%) | 0 (0%) | 1 (11%) |
| 25–29.9 | 55 (56%) | 51 (61%) | 9 (69%) | 6 (67%) |
| ≥30 | 40 (41%) | 29 (35%) | 4 (31%) | 2 (22%) |
| **Parity** | | | | |
| Primigravida | 50 (51%) | 43 (52%) | 9 (69%) | 5 (56%) |
| Multigravida | 46 (47%) | 38 (46%) | 3 (23%) | 4 (44%) |
| Unknown | 2 (2%) | 2 (2%) | 1 (8%) | 0 (0%) |
| **Number of sessions attended (data from SW)** | | | | |
| 10+ | 19 (19%) | 19 (23%) | 6 (46%) | 4 (44%) |
| 6–9 | 8 (8%) | 8 (10%) | 3 (23%) | 1 (11%) |
| 1–5 | 19 (19%) | 15 (18%) | 3 (23%) | 3 (33%) |
| None | 52 (53%) | 41 (49%) | 1 (8%) | 1 (11%) |

*Includes those who responded either to 6-month questionnaire only or to both 6-month and 12-month questionnaires.
BMI, body mass index; EGWG, excessive gestational weight gain.; SW, Slimming World.

not knowing she could take her baby with her (or not choosing to do so):

> I chose to breastfeed on demand so I would not have been able to attend any scheduled sessions easily. (Rose; gained 12 kg 12mth)

Two participants found the offer too soon postbirth; stating that 6 months postpartum would have been easier. Other 'opportunity' factors included their own or their infants ill health, or social factors including lack of confidence (capability) due to perceived language barriers reported by a non-native English speaker, and others stating they were leaving the country for an extended period, or leaving the city and not realising that they could attend groups elsewhere. Two participants experienced errors in registration that prevented them from enrolling in time. The location of the groups was only mentioned as a barrier by five women.

In addition to these 'opportunity' factors, there were 'motivational' barriers to uptake for some women who were either happy with their weight, stated they had already lost weight, or did not feel motivated to lose weight. Others did not believe SW to be suitable, either because they did not believe it promoted an effective or balanced diet, or due to a perceived lack of relevance for new mums:

> I was asked whether I would be comfortable with people cheering if I'd lost weight which made me think I had done something wrong rather than have a child! And also that the programme was targeted to teach me how to eat better—for example, by using olive oil and not eating ready meals—which I'd never done. So it sounded more like something for unhealthy people with bad habits, not new mums. (Lily; lost 6.5 kg 12mth)

Non-attendance was more prevalent among non-White British/Irish women. The reasons given for this were predominantly 'opportunity' issues regarding accessibility such as childcare/lack of fit with routine. Similar reasons were given by White British/Irish women who did not attend any sessions. Only two women who did

not attend any sessions commented on aspects of the diet presenting barriers (one White European background, and the other Asian).

### Retention

Although nearly half of women assigned to SW attended at least one session (46/98, 47%), only half of these (n=19/46) attended the full 'dose' (ie, 10+sessions); the mean number of sessions was 6.74 (SD 3.94). Virtually all women who only attended a few sessions attended these in consecutive weeks. Those who attended greater numbers of sessions (eg, 5–9 sessions) sometimes had 1 or 2 week gaps. Five key themes emerged which differed across the three groups (10+, 6–9 and 1–5 sessions) and help to explain barriers and facilitators to retention. Each is considered below and summarised in table 2 where they are mapped to behaviour change techniques (BCTs) that may mitigate barriers.

### Motivation: weight loss aspirations

Women attending 10+ sessions, and some women attending fewer sessions but losing weight over the data collection period, tended to perceive themselves as over-weight, to understand why they gained weight and to perceive weight loss as a priority.

> I guess I got into the routine of comfort eating and putting on more weight. I thought that if I didn't take this opportunity I would have really been at a loss. (Violet; 11; lost 17.5 kg 12mth)

In contrast, women attending 6–9 sessions and gaining weight, although acknowledging they were overweight and understanding why, were less likely to prioritise weight loss; they commonly said that adapting to life with their baby was more important:

> For me three months into having had a baby, I wasn't standing there thinking I've got to lose the weight. I was standing there thinking, wow how do I reacclima-tise and recalibrate my life to accommodate this little person. (Aster; 6; gained 2.8 kg 12mth)

Weight reduction was also given low priority by women attending 1–5 sessions, who were less likely to recognise they were overweight, and/or state they did not gain much weight during their pregnancy, as the rationale for not prioritising weight reduction, compared with women attending more sessions.

### Motivation (psychological capability): beliefs and expectations regarding weight loss

A common theme among women attending 10+ sessions was that the intervention had altered their beliefs about how to lose weight, particularly regarding the role of exercise:

> I've gone on various diets but half-heartedly, and I've always done it though exercise … So it's a kind of rev-elation that it's 90% food and 10% exercise which is what I learnt at Slimming World … seeing 5½ lbs loss

in the first week confirmed it would work. I am proud of myself. (Heather; 12; lost 2.6 kg 12mth)

Women who attended fewer SW sessions were more likely to question the intervention, often because they felt it failed to prioritise exercise as they had expected.

> I just felt there should also be an activity, there wasn't any exercise classes or anything like that and maybe that side of things didn't attract me. (Iris; 1; gained 4.4 kg 12mth)

Other beliefs that explained poor retention included that SW was unsuitable for breastfeeding mothers, or that their needs were broader than weight management (eg, also requiring postnatal mental-well-being support).

### Motivation/capability: understanding and implementing the intervention

A key 'capability' factor was the extent to which women understood and were able to implement the SW programme. Women who attended 10+ sessions described following the dietary plan and using SW online resources to support them. They were motivated by the health and well-being benefits they experienced for themselves and for their families:

> It felt to me to be sustainable, not a harsh diet that I was only going to do for a few weeks. I was still enjoy-ing myself and enjoying my food and I was still los-ing weight, so that was very motivating. (Hibiscus; 10; gained 0.3 kg 12mth)

In contrast, women who attended 6–9 sessions appeared to have a poorer understanding of the SW plan and/or reported difficulties adapting to the plan:

> It's quite a low sugar, low fat diet and I was reluctant to cut my fat levels too much so I tried to keep some full fat milk and I would still eat more fatty avocados. I just took the free food and went with it without actu-ally counting syns…I am not convinced that sticking to the plan would actually help me maintain longer term. (Lavender; 9; lost 0.4 kg 12mth)

Many gained weight and their motivation to attend may have been affected by lack of perceived benefit. For women who attended 1–5 sessions, these barriers were most apparent, suggesting that there might be a 'dose–response' effect: the more sessions women attended, the more likely it was that they would feel 'capable' and thereby 'motivated' to successfully imple-ment lifestyle changes.

### Social opportunity and motivation: social context

Opportunity factors relating to both the internal (inter-vention) and external (home support) social context were important explanatory factors for retention. Inter-nally, women who attended 10+ sessions were most likely to describe identifying with other SW group members, seeing their group consultants as helpful in adapting the programme to their circumstances (eg, breastfeeding)

**Table 2** Barriers to uptake and retention mapped to COM-B and to potential behaviour change techniques

| COM-B | Theme | 10+Slimming World sessions | <10 Slimming World sessions | Behaviour change techniques that could help mitigate barriers |
|---|---|---|---|---|
| Motivation | Weight loss aspirations | ▶ Self-perceived as overweight | ▶ Some acknowledged overweight and understood why but viewed weight management as less of a priority | Provide information (leaflet/brief consultation) at outset to help influence knowledge and understanding regarding weight loss and consequences of action/ no action |
| | | ▶ Understood why gained weight | ▶ Others did not self-perceive as overweight | · Identity (framing/reframing; incompatible beliefs; identity associated with changed behaviour) |
| | | ▶ Desired to lose weight after birth | | ▶ Comparison of outcomes (pros and cons; credible source; comparative imagining of future outcomes) |
| | | | | ▶ Goal setting (problem solving) |
| | | | | ▶ Natural consequences (information about health consequences; salience of consequences; information about social and environmental consequences) |
| Capability/ motivation | Beliefs and expectations about SW | ▶ SW and resulting weight loss helped change any pre-existing ideas about how to lose weight, esp re: role of exercise | ▶ Mistrustful of SW, in part as perceived as failing to prioritise exercise in line with pre-existing beliefs, seen as unsuitable for breastfeeding women, or felt gap in care for example, in relation to postnatal mental well-being | Leaflet/brief consultation (as above) to include: |
| | | | | ▶ Shaping knowledge (Information about antecedents e.g. provide reliable info regarding weight gain/ loss; reattribution, eg, elicit perceived causes of weight gain/how to lose weight and suggest alternatives) |
| | | | | ▶ Goals and Planning (goal setting: behaviour for example, to make clear at the outset that the SW programme considers the role of exercise but focusses on diet; that 10+sessions are needed for it to be most effective; Goal setting: outcome – make clear at the outset that the outcome goals are focused on weight management, but that weight loss may have other positive outcomes such as improved sense of well-being/mental health. |
| | | | | ▶ Comparison of outcomes (pros and cons: discuss pros and cons of exercise vs diet for weight loss) |
| Capability | Understanding and implementing SW | ▶ Understood dietary plan, found easy to follow, planned meals and used online resources | ▶ Did not appear to understand the dietary plan, only partial implementation, and experienced difficulties Making dietary adjustments | Development of a bespoke booklet to give to women at randomisation that summarises key elements of the programme (eg, some top tips/swaps; a sample 7-day menu plan); as well as optimising the following within the programme |
| | | ▶ Considered plan unrestrictive, sustainable and compatible with postnatal lifestyle | | Goals and planning (goal setting: behaviour; action planning) |
| | | ▶ Positive benefits for them and their families | | Feedback and monitoring (self-monitoring of behaviour, eg, complete food diary and get feedback from consultant) |
| | | | | Shaping Knowledge (instruction on how to perform a behaviour) |
| | | | | Repetition/Substitution (behaviour practice and rehearsal; graded tasks) |

Continued

**Table 2** Continued

| COM-B | Theme | 10+Slimming World sessions | <10 Slimming World sessions | Behaviour change techniques that could help mitigate barriers |
|---|---|---|---|---|
| Opportunity | Social context/accessibility | ▶ Identified with group members/social bonds | ▶ Did not bond/socially identify with group | In the leaflet/brief consultation provided at randomisation, women can be reminded that they can change groups and consultants if desired; incorporate anticipation of impact of the programme on the family/partner into brief consultation (eg, how to discuss with partner; a section in the leaflet for partner/family to read) |
| | | ▶ Positive relationships with group consultants, personalised support | ▶ Consultants unsympathetic to postnatal challenges | Social support (unspecified; practical; emotional) |
| | | ▶ Supported by partners, childcare/sharing SW meals | ▶ Partner lack of support/partial adaptation to previous meals | Reward and threat (social incentive: explore whether their family/peers will congratulate them if they lose weight/attend SW sessions; restructure the social environment for example, by involving those who will be supportive of them attending SW, of dietary changes and weight loss) |

COM-B, capability, opportunity, motivation and behaviour; SW, Slimming World.

and generally perceived the groups as a supportive and safe environment:

> I was positively surprised how much the group was supportive and I'm also new in the area so it was a really good way to meet people, like meet new mums. I keep bumping into members in the supermarket and in the park exercising …. I felt part of a little community. (Orchid; 10; lost 3.7 kg 12mth)

> Consultant very positive, made baby welcome and was supportive. (Peony; lost 3.1 kg 12mth)

Those who attended <10 sessions were more likely to express negative views about their groups or consultants. Several mentioned having difficulty bonding or socially identifying with other group members, or that their consultants were unsympathetic to their specific postnatal challenges:

> The atmosphere was very city/work based and not welcoming to new mothers. (Poinsettia; 1; gained 1.8 kg 6mth)

> The woman who ran the group, wasn't very nice, she was very critical…. (Hyacinth; 7; gained 13.7 kg 12mth)

In relation to 'external' factors, women attending 10+sessions commonly reported benefitting from support from their partners, either practically (eg, childcare whilst they attended), or willingness to adapt their diets alongside their partner. One husband also joined SW: 'I took my husband with me. He lost quite a lot of weight as well. So it was good.' (Orchid; 10; lost 3.7 kg 12mth)

For others, the lack of partners' support presented a barrier to participating and/or losing weight:

> It's a lot easier to do (follow the plan) on the nights he (husband) is working. I suppose he just likes certain food … he likes to have a proper meal. (Hibiscus; 10; gained 0.3 kg 12mth)

> Husband not supportive. (Poinsettia; 1; gained 1.8 kg 6mth)

### Physical opportunity: accessibility of the intervention

Most women (35/46, 76%) stayed at the same group for all their sessions, the majority attending groups on weekdays (40/46, 87%) though the time of day was split fairly equally between morning and evening attendance. A key opportunity factor was the accessibility of the intervention, which either facilitated or created a barrier to attendance. Issues included *timing of onset* of the intervention, *ease of attendance* and *duration of the intervention.* Women had diverse views that were unrelated to the number of sessions attended or their weight change. In relation to intervention onset, most reported that starting SW between 8 and 16 weeks postbirth was acceptable. However, some felt it was too early in the postnatal period, coinciding with the time they might be experiencing difficulties with breastfeeding, concerns over their baby's health and the general transition to parenthood:

> couldn't concentrate while had baby with me as breastfeeding … offer too early in postnatal period, bad timing of groups with baby, exhausted. (Primrose; 2; lost 0.7 kg 12mth)

Regarding ease of attendance, most women could access an SW group that was located and timed to suit

them. However, two women who attended <10 sessions, stated having difficulties in this regard:

> That (group) was the only one within walking distance from my house…On most days; I didn't stay (after being weighed)…because of the inconvenient timing she was by that point in time she would be really ratty and ready for sleep. (Hyacinth; 7; gained 13.7 kg 12mth)

Other 'opportunity' reasons for missing sessions and/or not completing the programme included holiday plans, illness (in woman or infant) and childcare difficulties (for siblings or where they did not want to take their babies with them).

The third key accessibility issue concerned the intervention duration. Many who completed the programme considered it to be too short:

> I don't think 12 weeks is long enough to see any real benefits and to establish healthy eating/lose weight long term. I would say 6 months is optimal. (Tulip; 6; gained 3.6 kg 12mth).

Women could continue beyond 12 weeks but fees were applicable. Some women did this, for others the timing of the intervention ceasing presented a major barrier as coincided with the end of maternity benefit:

> Once the 12 weeks were up … that was the time when I (had) no money, so £5 a week ended up being quite a lot of money that I couldn't really afford … if it was possible to do it for a year (during) maternity leave that would be fantastic. (Azalea; 11; lost −1.8 kg 12mth)

None of the women who responded to questionnaire/s or participated in interview(s) mentioned cost as a facilitator to uptake or retention. White British/Irish women were more likely to attend 10+ sessions than women from other ethnic groups. Examination of the barriers between these groups indicated that women from non-White British/Irish groups were more likely to mention accessibility opportunity factors such as time, access and childcare as barriers to retention, as opposed to the White British/Irish women who were more likely to mention social context opportunity factors such as the consultant or group identity. We found no clear relationship between socioeconomic variables (household income or IMD decile) and uptake or retention, though all negative comments regarding the diet (n=5) were from women in the highest income groups.

## DISCUSSION

This is the first study to use behaviour change theory to explore uptake and retention in a postnatal weight management intervention. A number of modifiable barriers to uptake and retention were identified. Fewer than half of the intervention arm women attended at least one session; a rate consistent with weight management studies in

pregnancy[23 24] and general population.[25] Reported barriers could potentially be addressed using BCTs (see table 2)—for example through strengthening motivation to lose weight by tackling incompatible beliefs and/or supporting consideration of the pros and cons of weight loss and providing evidence regarding health consequences. Previous studies of weight management in pregnancy have highlighted the difficulty that midwives and other health professionals have discussing weight with women,[26–28] and indicate a training need to support effective conversations. Some women also found the timing of the intervention challenging: offering a wider commencement window (eg, allowing women to start SW anytime up to 6 months postbirth) may have increased uptake.

Only 19% of intervention arm women received the 'full dose' of 10+ sessions. Retention may be improved by offering enhanced support to help women adapt the dietary plan to their lifestyle and family needs, to address capability issues (eg, understanding and/or implementing the diet); and opportunity factors might be mitigated by ensuring women know there is flexibility regarding the group they attend; and by encouraging women to identify those in their social environment who will support their dietary changes and weight loss aspirations. This enhanced support may need to be tailored for different ethnic groups. There appeared to be a 'dose–response' relationship with women attending 10+ sessions reporting the greatest benefit. This concurs with findings from a general population study where those attending at least 75% of the offered sessions (also provided by SW) achieved the greatest weight loss.[19] Women who did not receive the full dose often had erroneous perceptions about the programme and its effectiveness or compatibility with their postnatal status, which in turn reinforced their decision to give up. The use of techniques to shape knowledge regarding effective weight loss and the SW programme at the time of study recruitment may mitigate this barrier and more closely align their expectations with the programme. As well as widening the commencement window, findings suggest that a longer duration of intervention may improve the sustainability of weight loss, particularly as the current duration ended at a similar time to women's statutory maternity leave payments ending. Recent evidence from an RCT comparing weight loss in adults with obese BMIs allocated to brief (booklet only), 12 week or 52 week programmes found the 52 week programme was both more clinically and cost effective than shorter programmes.[29]

To our knowledge, this is the first study that has used behaviour change theory to understand uptake and retention of postnatal women in a weight management programme, and included a diverse inner-city population with women with various levels of attendance. The COM-B was originally developed to inform intervention development, though has been applied to evaluation of interventions in various settings.[30–33] Its use in this feasibility study enables both evaluation and design,

and findings will inform changes to enhance uptake and retention for the intervention in the main study. The use of framework analysis ensured data analysis was transparent and enabled all research team members to contribute. While the interview sample was small, the integration of the survey and interview findings is a strength of this study. While only a small number of women who did not take up the intervention were interviewed, 78% (42/51) of those who did not attend completed a survey where they explained why. The interviews were via telephone rather than face-to-face and this is a potential limitation. However, all the women said that they preferred telephone to face-to-face. Moreover, we consider that telephone interviews were appropriate as the topic of the interviews were not covering sensitive topics that might especially warrant face-to-face, nor was it seen as necessary to gather contextual information about the participant's environment. It is not surprising that women preferred telephone interviews as this is a practical option when caring for a young infant. The findings regarding the potential impact of ethnicity on uptake and retention require further exploration in a larger study. As a single-centre feasibility study, findings cannot be generalised, though the intervention is commercially available throughout the UK and in several countries worldwide, which may enhance the relevance of findings. The analysis and interpretation of uptake and retention is limited to that which might optimise the intervention for a future trial. Discussions with women with BMI $\geq 25\,kg/m^2$ who had recently used maternity services in the local area informed the choice of intervention: they had a clear preference for non-NHS setting and groups that were not solely for women with babies. However, evidence from this study suggests that some women may have preferred NHS settings and groups focussing on women with infants. Findings will be used to optimise the potency of the intervention in a future trial, with an embedded process evaluation to further evaluate the impact of proposed BCTs on uptake and retention.

**Author affiliations**
[1]School of Health Sciences, University of Surrey, Guildford, UK
[2]National Childbirth Trust, London, UK
[3]Population Health Research Institute, St George's University of London, London, UK
[4]Institute of Social Marketing and Health, University of Stirling, Stirling, UK
[5]Warwick Clinical Trials Unit, Warwick Medical School, University of Warwick, Coventry, UK

**Acknowledgements** We thank the women who participated in this study, and Sheila O'Connor and Victoria Craig (Research Midwives) for supporting recruitment and data collection. We would also like to thank members of our trial steering group.

**Collaborators** SWAN trial team members: Dr Amanda Avery, University of Nottingham Dr Andy Healey, King's College London Ms Nina Khazaededah, Guy's and St Thomas' NHS Foundation Trust Dr Sara Mc Mullen, National Child birth Trust Professor Eugene Oteng-Ntim, King's College London Dr Bimpe Oki, Lambeth and Southwark Public Health Professor Lucilla Poston, King's College London Mr Paul Seed, King's College London.

**Contributors** CT and DB developed the study concept and study design, with the support of MU. VB completed data collection and VB, CT and MZ were responsible for data analysis. All authors contributed to interpretation of findings. DB had primary responsibility for overseeing the overall SWAN trial. CT drafted the first version of the manuscript, with the support of MU, MZ and DB. All authors read and approved the final manuscript.

**Funding** This study is funded by the National Institute for Health Research (NIHR) Public Health Research Programme (ref no. 14/67/14). The views expressed are those of the author(s) and not necessarily those of the NIHR or the Department of Health and Social Care.

**Competing interests** AA (a member of the SWAN trial team), alongside her academic position at the University of Nottingham, also holds a consultancy position at Slimming World (Alfreton, UK). Neither AA nor Slimming World (Alfreton, UK) had access to the study data or were involved in data collection or analyses. DB was supported by the National Institute for Health Research (NIHR) Collaboration for Leadership in Applied Health Research and Care South London (NIHR CLAHRC South London) at King's College Hospital NHS Foundation Trust. The other authors declare that they have no competing interests in relation to this study.

**Patient consent for publication** Not required.

**Ethics approval** Ethics approval was granted by the Health Research Authority London—Camberwell St Giles REC (16/LO/1422). Women who met the purposive criteria were approached initially by Research Midwives who provided a study information sheet and consent form (either in face-to-face contacts or by post). Consent forms were either returned in person to the Research Midwives or posted directly to the researcher. Those who consented were then contacted by the researcher to answer questions, reconfirm consent verbally and arrange interviews. Consent was also reconfirmed verbally at the start of interviews.

**Provenance and peer review** Not commissioned; externally peer reviewed.

**Data availability statement** Data are available upon reasonable request. The data sets generated and/or analysed as part of this study are not publicly available due to them containing information that could compromise research participant privacy/consent but are available from the corresponding author on reasonable request.

**ORCID iDs**
Cath Taylor http://orcid.org/0000-0001-6239-4744
Debra Bick http://orcid.org/0000-0002-8557-7276

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
