## [Reviewer comments · BMJ Open]

ARTICLE DETAILS

TITLE (PROVISIONAL)	Barriers and facilitators to uptake and retention of inner-city ethnically diverse women in a postnatal weight management intervention: a mixed methods process evaluation within a feasibility trial in England
AUTHORS	Taylor, Cath; Bhavnani, Vanita; Zasada, Magdalena; Ussher, Michael; Bick, Debra

VERSION 1 – REVIEW

REVIEWER	Louise Goff King's College London, UK
REVIEW RETURNED	24-Oct-2019

GENERAL COMMENTS	In this manuscript the authors report the use of the COM-B model for evaluating barriers and facilitators to uptake and retention of women in a post-partum weight loss intervention. Whilst the authors point out the novelty of using the COM-B model to evaluate uptake and retention in a weight loss intervention, I would urge them to give more consideration of the limitations of this approach and their findings. There is only half a sentence in the final section of the discussion about the limitations of their work (i.e. generalisability). Please also consider the limitations of applying the COM-B model, which is intended more for the design of behavioural interventions, to evaluating uptake and retention, as well as other limitations like the relatively small number of interviews that were conducted with women who did not take up the intervention. The interviewees were invited according to attendance, weight loss and ethnicity. Were there any themes in the findings according to the ethnicity of the women? In the methodology, please can the authors provide the dates over which the study was conducted, and also provide details for the slimming world sessions, were they daytime or evening delivery? Citation number 15 is important but is incomplete, please provide full and correct details. The themes relating to uptake are all barriers. Were there no facilitators identified? It is particularly important to consider barriers and facilitators that relate to the research process versus those that relate to the slimming world intervention. Interestingly cost came up to a barrier to continuing to attend SW beyond the free sessions but there is no mention of whether it was a facilitator to participation.
---

	A minor point - the grammar in the section of the results on 'motivation: weight loss aspirations' is incorrect, this section needs rewriting. I have some concerns about the depth of the information that was gathered, particularly reading some of the quotes that are included in the manuscript, which appear to be quotes from the questionnaires rather than interviews e.g. 'husband not supportive'. I would have expected the interviews to have yielded greater depth of information. The recommendations provided in Table 2 of behaviour change techniques that could help mitigate barriers are pretty basic, probably because the authors are limiting their thinking to optimising the current intervention for a future trial. It would be nice to see the authors push their thinking a little further and make suggestions for interventions beyond the SW model e.g. was there evidence to suggest that a weight loss group specifically for new mothers, involving exercise and providing childcare would be favoured?
--	---

REVIEWER	Louise Hayes Newcastle University, UK
REVIEW RETURNED	07-Jan-2020

GENERAL COMMENTS	Thank you for the opportunity to review this interesting manuscript. I commend the authors for conducting this process review. All too often trials of interventions in pregnancy and the post-natal period have been unsuccessful and little has been done to understand why. On the whole I found this paper interesting and the results are consistent with what is already known about the barriers and facilitators to pregnant and post-natal women engaging with lifestyle interventions to promote weight loss. It is helpful to have the addition of suggestions, grounded in behaviour change theory and supported by behaviour change techniques, of ways in which barriers could be addressed. As the authors themselves point out the study was undertaken in an ethnically and economically diverse population. However, the analysis was restricted to a comparison between those who attended at least 10 or fewer than 10 SW sessions. It would be interesting to compare between women of different ethnic and socio-economic background. It seems likely that different barriers (and therefore approaches to address them) would be needed (and would need to be tested) in different groups of women. It is notable that attrition (in terms of survey response and 6 and 12 month interview) was greater in women from non-White ethnic groups. In terms of the classification into 10+ vs <10 sessions it would also be interesting to explore if those attending fewer sessions tended to attend those sessions initially and then drop out or to attend fewer sessions across the 12 week period (i.e. did they attend 4 sessions for the first 4 weeks and then stop attending or attend one session every 3 weeks across the whole period). I have a few minor comments:  1. In the abstract I found the sentence '....opportunity issues, particularly in relation to the social context of the group....' to be
---

	confusing. I wouldn't call the social context of the group an opportunity issue. 2. In the introduction the sentence '...infants are at risk of higher BMI and blood pressure....' relates to obese pregnancy, not postnatal weight retention as is implied by the way this paragraph is currently written. I suggest the focus here should be on the risks of postnatal weight retention for a subsequent pregnancy and the impact of this on offspring. 3. Reference number 15 is not provided. 4. In methods there is a typo in the participants section - should read 'with BMIs > 25 (i.e. greater than or equal to). 5. In methods in the 'Questionnaire component' section it says that measures of diet etc are reported elsewhere, but no reference is given. 6. In methods in the 'Interview component' section it is reported that all interviews were by telephone, but any relevance or importance of this in terms of the findings is not discussed. 7. In results under 'Uptake' it is noted that some women reported 'leaving London' - I just wondered if the reference to London should be removed as previously the population was described as 'inner-city', as if care had been taken not to disclose the location.
--	--

VERSION 1 – AUTHOR RESPONSE

Reviewer 1	
Whilst the authors point out the novelty of using the COM-B model to evaluate uptake and retention in a weight loss intervention, I would urge them to give more consideration of the limitations of this approach and their findings. There is only half a sentence in the final section of the discussion about the limitations of their work (i.e. generalisability). Please also consider the limitations of applying the COM-B model, which is intended more for the design of behavioural interventions, to evaluating uptake and retention, as well as other limitations like the relatively small number of interviews that were conducted with women who did not take up the intervention.	Thank you for this comment. Whilst we agree that the COM-B was originally intended to inform intervention design, it (and the theoretical domains framework associated with the COM-B) has increasingly been used to evaluate interventions (e.g. use in systematic reviews evaluating behaviour change interventions [30-33], and was used in a large study of uptake and retention of weight management in general populations (Reference 14 cited in the introduction). As this is a feasibility study with scope to adapt to enhance the intervention for a main study we would also argue that this is being used with intervention development in mind and thereby do not see this as a limitation. We have added the following sentence to the discussion regarding this (p14): “The COM-B was originally developed to inform intervention development, though has been applied to evaluation of interventions in various settings [30-33]. Its use in this feasibility study enables both evaluation and design, and findings will inform changes to enhance uptake and retention for the intervention in the main study”.

	Whilst we only interviewed a small number of women who did not take up the intervention (sampling was informed by guidance for feasibility studies, Ref 20), the investigation of barriers and facilitators to uptake and retention was mixed methods and as such the analysis explores the reasons given by women in their survey responses concerning why they didn't go to Slimming World. 41 of the 52 women that did not attend any Slimming World sessions responded to the survey (79%). We have added the following sentence to the discussion (p14): “Whilst the interview sample was small, the integration of the survey and interview findings is a strength of this study. Whilst only a small number of women who did not take up the intervention were interviewed, 78% (42/51) of those that didn't attend completed a survey where they explained why”.
The interviewees were invited according to attendance, weight loss and ethnicity. Were there any themes in the findings according to the ethnicity of the women?	Thank you for this question. Although interview sampling aimed to capture diversity according to ethnicity (and did achieve this), for this paper the analysis examined reasons for different rates of attendance (e.g. comparing non-attenders, those who attended less than, and more than ten sessions). The interview sample was designed in line with feasibility goals [Ref 20 in the paper] and provides insufficient ethnic diversity within the three different attendance groups to be able to draw any robust conclusions. However, we have examined the data as a whole (including questionnaire responses) according to ethnicity, though for uptake only three White British/Irish women who did not attend any sessions completed the questionnaire therefore we do not have confidence in comparing reasons for non-uptake by ethnicity. More non-white women did not attend any sessions however, and the reasons given for this were predominantly 'opportunity' issues regarding accessibility, such as childcare/lack of fit with routine. Similar reasons were given by the white women who did not attend. Only two women that did not attend any sessions commented on aspects of the diet presenting barriers, and one was from a white European background and the other Asian.

	We have added the following to the manuscript to present this finding (p8): “Non-attendance was more prevalent amongst non-White British/Irish women. The reasons given for this were predominantly ‘opportunity’ issues regarding accessibility such as childcare/lack of fit with routine. Similar reasons were given by White British/Irish women who did not attend any sessions. Only two women that did not attend any sessions commented on aspects of the diet presenting barriers (one White European background, and the other Asian).” In relation to retention, White British/Irish women were more likely to attend 10+ sessions than women from other ethnic groups. Examination of the barriers between these groups indicated that women from non-White British/Irish groups were more likely to mention accessibility/opportunity factors such as time, access, childcare as barriers to retention, as opposed to the White British/Irish women who were more likely to mention social context/opportunity factors such as the consultant or group identity. We have added the paragraph above to the manuscript (p12). We have also added two mentions regarding the potential impact of ethnicity on uptake and retention in the discussion: “This enhanced support may need to be tailored for different ethnic groups”(p13). and “The findings regarding the potential impact of ethnicity on uptake and retention require further exploration in a larger study” (p14)
In the methodology, please can the authors provide the dates over which the study was conducted, and also provide details for the slimming world sessions, were they daytime or evening delivery?	The study was conducted between 01/12/2015 – 30/11/2018. This detail has been added to the manuscript (p4).

	Women were able to attend any sessions that suited them. Sessions ran in morning, afternoon and evening and across all days of the week. We have added the following summary sentence to the manuscript (p11): “Most women (35/46, 76%) stayed at the same group for all their sessions, the majority attending groups on weekdays (40/46, 87%) though the time of day was split fairly equally between morning and evening attendance”.
Citation number 15 is important but is incomplete, please provide full and correct details.	Apologies this has now been corrected and replaced with the reference for the main study paper which has now been published (see response to Reviewer 1 regarding this for more detail).
The themes relating to uptake are all barriers. Were there no facilitators identified? It is particularly important to consider barriers and facilitators that relate to the research process versus those that relate to the slimming world intervention. Interestingly cost came up to a barrier to continuing to attend SW beyond the free sessions but there is no mention of whether it was a facilitator to participation.	Thank you for this comment. The facilitators to uptake had not been reported explicitly as they were subsumed in the reasons that explained retention. The key facilitators reported by women to be important at supporting them with initial uptake were opportunity: it being the right post-natal timing and fitting with their post-natal routine; and motivation: having aspiration for weight loss. All of these factors were also barriers for others as reported in the paper. In relation to cost, none of the women who responded to questionnaire/s or participated in interview(s) mentioned cost as a facilitator to uptake or retention, other than those who mentioned it as barrier for continuing. We have added a sentence to the section in the results that mentions cost to state (p12): “None of the women who responded to questionnaire/s or participated in interview(s) mentioned cost as a facilitator to uptake or retention” We agree the barriers and facilitators that relate to the study participation are separate to those that relate to the intervention. This paper is focussed on the barriers and facilitators for the intervention (the study related factors are reported in the

	feasibility trial paper [15]. We have added the following to the methods section (p6): “Barriers and facilitators to study participation were considered separately to those that relate to the intervention and are reported elsewhere [15].”
A minor point - the grammar in the section of the results on 'motivation: weight loss aspirations' is incorrect, this section needs rewriting.	We have reviewed and revised this section (p9).
I have some concerns about the depth of the information that was gathered, particularly reading some of the quotes that are included in the manuscript, which appear to be quotes from the questionnaires rather than interviews e.g. 'husband not supportive'. I would have expected the interviews to have yielded greater depth of information.	The data presented include both questionnaire and interview data – as explained in the methods. Quotations have been taken from both sources. Interviews did yield depth of information and the other quotation taken from this section was from an interview (Hibiscus), however “husband not supportive” was taken from a woman’s questionnaire response to the question asking why she did not continue with the groups and no further explanation was provided.
The recommendations provided in Table 2 of behaviour change techniques that could help mitigate barriers are pretty basic, probably because the authors are limiting their thinking to optimising the current intervention for a future trial. It would be nice to see the authors push their thinking a little further and make suggestions for interventions beyond the SW model e.g. was there evidence to suggest that a weight loss group specifically for new mothers, involving exercise and providing childcare would be favoured?	Thank you for this comment. We acknowledge and agree that we have limited our thinking to optimising the current intervention for a future trial, as the study was conducted for this purpose. In relation to the reviewer’s suggestions about evidence to suggest other models may be favoured, the choice of intervention was in itself informed by discussions with local PPI groups which highlighted that women did not want support from NHS staff or from groups which only targeted women with babies. Clearly there will never be a one-size fits all though and evidence from some women who participated in this study suggests they may have favoured an intervention that was specifically for new mothers, explicitly involving doing exercise and/or providing childcare. We have added the following to the discussion in the manuscript (p14): “The analysis and interpretation of uptake and retention is limited to that which might optimise the intervention for a future trial. Discussions with women with BMI ≥ 25 kg/m² who had recently used maternity services in the local area informed the choice of intervention: they had a clear preference for non-NHS setting and groups that were not

	solely for women with babies. However, evidence from this study suggests that some women may have preferred NHS settings and groups focusing on women with infants”.
Reviewer 2	
Thank you for the opportunity to review this interesting manuscript. I commend the authors for conducting this process review. All too often trials of interventions in pregnancy and the post-natal period have been unsuccessful and little has been done to understand why. On the whole I found this paper interesting and the results are consistent with what is already known about the barriers and facilitators to pregnant and post-natal women engaging with lifestyle interventions to promote weight loss. It is helpful to have the addition of suggestions, grounded in behaviour change theory and supported by behaviour change techniques, of ways in which barriers could be addressed.	Thank you for these positive and supportive comments.
As the authors themselves point out the study was undertaken in an ethnically and economically diverse population. However, the analysis was restricted to a comparison between those who attended at least 10 or fewer than 10 SW sessions. It would be interesting to compare between women of different ethnic and socio-economic background. It seems likely that different barriers (and therefore approaches to address them) would be needed (and would need to be tested) in different groups of women. It is notable that attrition (in terms of survey response and 6 and 12 month interview) was greater in women from non-White ethnic groups.	Please see response to Reviewer 1 regarding ethnicity. In relation to socioeconomic background, we examined the reasons for uptake and retention according to both women’s household income and IMD decile. There were no clear patterns in relation to either of these variables for uptake or for retention, though we did find that all of the negative comments regarding the diet (from 5 women) came from women in the highest income groups. We have added the following sentence to the results to share this finding: “We found no clear relationship between socio-economic variables (household income or IMD decile) and uptake or retention, though all negative comments regarding the diet (n=5) were from women in the highest income groups”.
In terms of the classification into 10+ vs <10 sessions it would also be interesting to explore if those attending fewer sessions tended to attend those sessions initially and then drop out or to attend fewer sessions	Thank you for this important question. We have re-examined the data in relation to this, and can confirm that virtually all of the women who dropped out after a few sessions attended these in consecutive weeks. Those who attended greater

across the 12 week period (i.e. did they attend 4 sessions for the first 4 weeks and then stop attending or attend one session every 3 weeks across the whole period).	numbers of sessions (e.g. 5-9 sessions) sometimes had one or two week gaps. We have added the following to the manuscript: “Virtually all women who only attended a few sessions attended these in consecutive weeks. Those who attended greater numbers of sessions (e.g. 5-9 sessions) sometimes had one or two week gaps.”
I have a few minor comments: 1. In the abstract I found the sentence '....opportunity issues, particularly in relation to the social context of the group....' to be confusing. I wouldn't call the social context of the group an opportunity issue.	Thank you. Opportunity in relation to the COM-B refers to both physical and social opportunity and this refers to the finding that it was the social opportunity factors that were particularly relevant. We have reworded the sentence to make this clearer: “ ... as were social opportunity issues, particularly in relation to factors such as the extent of group identity and relationship with the group consultant; and physical opportunity such as perceived support from and fit with family lifestyle”
2. In the introduction the sentence '....infants are at risk of higher BMI and blood pressure....' relates to obese pregnancy, not postnatal weight retention as is implied by the way this paragraph is currently written. I suggest the focus here should be on the risks of postnatal weight retention for a subsequent pregnancy and the impact of this on offspring.	Thank you for opportunity to clarify this. We have reworded this and separated it from the previous sentence to say (p3): “The infants of women with obese BMIs in pregnancy are also at risk of higher BMI and blood pressure themselves in childhood and young adulthood”
3. Reference number 15 is not provided.	Apologies, this has now been provided.
4. In methods there is a typo in the participants section - should read 'with BMIs > 25 (i.e. greater than or equal to).	Thank you for pointing this out, it has now been corrected.
5. In methods in the 'Questionnaire component' section it says that measures of diet etc are reported elsewhere, but no reference is given.	Apologies this has now been corrected – and now references 15 (the main trial paper).
6. In methods in the 'Interview component' section it is reported that all interviews were by telephone, but any relevance or importance of this in terms of the findings is not discussed.	Thank you. All women were offered either face to face or telephone interviews, and all chose to participate via telephone. We have added the following to the discussion (p14): “The interviews were via telephone rather than face-to-face and this is a potential limitation. However, all the women said that they preferred

	telephone to face-to-face. Moreover, we consider that telephone interviews were appropriate as the topic of the interviews were not covering sensitive topics that might especially warrant face-to-face, nor was it seen as necessary to gather contextual information about the participant's environment. It is not surprising that women preferred telephone interviews as this is a practical option when caring for a young infant."
7. In results under 'Uptake' it is noted that some women reported 'leaving London' - I just wondered if the reference to London should be removed as previously the population was described as 'inner-city', as if care had been taken not to disclose the location.	Thank you for pointing this out. We have changed London to 'the city' (p8).

VERSION 2 – REVIEW

REVIEWER	Louise Hayes Newcastle University, UK
REVIEW RETURNED	27-Apr-2020
GENERAL COMMENTS	The reviewer comments on this manuscript have been dealt with satisfactorily and I am happy to recommend it for publication.

VERSION 2 – AUTHOR RESPONSE

In relation to the editorial requests, we have detailed the changes we have made below, and have uploaded marked and unmarked versions of the manuscript as requested:

1. While we understand that pseudonyms have been used to represent each of the participant quotes used in the results section, some of the pseudonyms used are common names. To save any confusion in this regard, please put a disclaimer in the results section that states that pseudonyms have been used.

Response: We had inserted a sentence to this effect in the methods (at the end of page 6), and have replicated this sentence at the start of the Results section (page 7) as well. Please can you confirm that you wish to have it in both places?

2. Please include information regarding consent alongside the ethics statement in the manuscript.

Response: We had included the details regarding approach and informed consent within the methods (first paragraph page 6) and have copied this information also to be placed in the middle of page 7 with the ethics approval information. Please can you confirm you wish to have this in both places?